# Risk Factors and Spatiotemporal Analysis of Classical Swine Fever in Ecuador

**DOI:** 10.3390/v15020288

**Published:** 2023-01-19

**Authors:** Alfredo Acosta, Klaas Dietze, Oswaldo Baquero, Germana Vizzotto Osowski, Christian Imbacuan, Alexandra Burbano, Fernando Ferreira, Klaus Depner

**Affiliations:** 1Institute of International Animal Health/One Health, Friedrich-Loeffler-Institut, 17493 Greifswald, Germany; 2Laboratory of Epidemiology and Biostatistics, School of Veterinary Medicine and Animal Science, Preventive Veterinary Medicine Department, University of São Paulo, São Paulo 05508-270, Brazil; 3General Coordination of Animal Health, Phyto-Zoosanitary Regulation and Control Agency, Quito 170903, Ecuador

**Keywords:** risk factors, classical swine fever, spatiotemporal, risk-based surveillance, case-control, vaccination, Ecuador

## Abstract

Classical swine fever (CSF) is one of the most important re-emergent swine diseases worldwide. Despite concerted control efforts in the Andean countries, the disease remains endemic in several areas, limiting production and trade opportunities. In this study, we aimed to determine the risk factors and spatiotemporal implications associated with CSF in Ecuador. We analysed passive surveillance and vaccination campaign datasets from 2014 to 2020; Then, we structured a herd-level case–control study using a logistic and spatiotemporal Bayesian model. The results showed that the risk factors that increased the odds of CSF occurrence were the following: swill feeding (OR 8.53), time until notification (OR 2.44), introduction of new pigs during last month (OR 2.01) and lack of vaccination against CSF (OR 1.82). The spatiotemporal model showed that vaccination reduces the risk by 33%. According to the priority index, the intervention should focus on *Morona Santiago* and *Los Rios* provinces. In conclusion, the results highlight the complexity of the CSF control programs, the importance to improve the overall surveillance system and the need to inform decision-makers and stakeholders.

## 1. Introduction

With the growing demand for animal protein in the Andean region, especially pork, diseases such as classical swine fever (CSF) are gaining importance as they are limiting local production and potential export opportunities for affected countries. The per capita consumption of pork has increased in Ecuador from 6.88 kg in 2009 to 10.9 kg in 2018 (www.aspe.org.ec, accessed on 1 December 2022). The overall importance of pig production is related not only to meat consumption but also to cultural traditions. In Andean communities, pigs play a central role as a source of protein, festivities and savings [1]. 

CSF is considered a relevant re-emerging viral disease of pigs, caused by Pestivirus of the family *Flaviviridae* [2]. The only natural host are domestic and wild pigs (*Suidae* family). Clinical signs are variable and depend on the viral strain, host immune response, age, general health status and concomitant infections [3].

The course of the disease includes acute, chronic and persistent forms according to their duration [4,5]. Transmission occurs mainly by direct contact between infected and susceptible animals via the oronasal route but also indirectly through contact with contaminated clothes, vehicles, equipment, and ingestion of contaminated and undercooked meat, e.g., as part of swill feeding [6]. Outbreaks of CSF usually have dramatic consequences. Control measures include long quarantine periods, movement restrictions, emergency vaccination, or culling of the pigs. Additionally, major impacts on animal welfare occur [7,8]. For instance, the 1998 epidemic in the Netherlands had an estimated cost of 2.3 billion US dollars and about 10 million pigs were destroyed [9]. Countries with endemic status are banned from export, and therefore the impact of the disease on the economy and public health worldwide is high. In Ecuador, the economic impact caused by CSF was estimated by the National Veterinary Service (NVS) to be USD 6 million per year. Since many affected farms were low-income backyard producers, the impact of CSF on them is substantial [10].

In South America, the disease is considered endemic in Guyana, Suriname, the North and Northeast regions of Brazil and the Andean Community, and these regions struggle to implement successful control programs [11,12].

The CSF eradication Project in Ecuador started in 2012; the first national vaccination campaign was gradually launched in 2014 with a locally produced lapinised Chinese vaccine strain [13]. The highest coverage was achieved in 2019 (2.7 million doses) due to a compulsory vaccination campaign, government subsidies and coordination with stakeholders (commercial and industrial producers’ associations). However, the field response and data analysis capacity of the veterinary service was limited, and in 2022, the disease was still present (https://wahis.oie.int, accessed on 1 December 2022). In this regard, the NVS planned to enhance their analysis capacity and apply risk-based surveillance. One of the main challenges in applying risk-based surveillance is to identify the factors associated with the occurrence of CSF [14,15,16,17,18,19]; in developing countries, due to very particular production systems, the risk factors may be different. Recently, for some countries in South America such as Colombia [20], Brazil [21], and Peru [22], this issue has been addressed.

For Ecuador despite the importance and need for local CSF risk factors, little is still known concerning control measures and public policy. This is the first time that official data have been analysed in this regard. 

The objectives of this study were to determine the risk factors associated with the occurrence of CSF and to analyse the spatiotemporal implications in order to identify the most at risk locations.

## 2. Materials and Methods

### 2.1. Datasets

Data were collected by the NVS from January 2014 to November 2020 in mainland Ecuador, excluding the Galapagos Islands, as they are a recognised CSF-free zone [2]. The information was stored in two databases: (1) Ecuador’s animal health information system (SIZSE) created to record paper questionnaires for notifiable diseases from passive surveillance since 2014 (https://sistemas.agrocalidad.gob.ec/sizse/, accessed on 1 January 2021); (2) the Unified information manager (GUIA) developed by the NVS to manage cadastre, mass vaccination campaigns against CSF, and movements since 2016 (https://guia.agrocalidad.gob.ec/agrodb/ingreso.php, accessed on 1 January 2021). Shapefiles of administrative units of Ecuador were downloaded from the Institute of Statistics and Census (INEC) (https://www.ecuadorencifras.gob.ec/division-politico-administrativa/, accessed on 1 February 2021). 

All raw data were then imported and processed with R version 4.2.1 (https://CRAN.R-project.org/, accessed on 1 January 2022).

### 2.2. Surveillance, Factors Influencing the Risk and Case Definition

Surveillance of swine diseases in Ecuador is the responsibility of the SNV. Producers and citizens notify suspect animals to the local veterinary service using mainly telephone, then official veterinarians investigate the suspect animals, as well as perform clinical anamnesis, conduct epidemiological investigations, collect the necessary samples and fill the information in the questionnaire; after laboratory diagnosis, when the suspect event is ruled out or when a positive case is confirmed, official control measures are taken which include quarantine, culling, movement restrictions and ring vaccination in the area.

Using the information gathered by the passive surveillance system, a case-control study was structured to identify factors associated with CSF occurrence. The retrospective analysis used the variables collected historically by the surveillance system and the vaccination campaign. The databases were merged using the individual identification of the owner of the premises.

The variables included in the analysis were selected based on biological plausibility, published literature, and considering their association with the occurrence of classical swine fever. [15,23,24]. Subsequently, they were organised considering risk characteristics according to the RiskSur Surveillance design framework (www.fp7-risksur.eu, accessed on 1 March 2021) [25] and grouped into population level, herd level and animal level.

Laboratory testing was performed at the National Reference Laboratory (headquarters in Quito). Virus detection was carried out by a commercially available antigen ELISA screening test (PrioCheck^®^ CSFV), based on the double antibody sandwich (DAS) principle with a sensitivity of 97% and a specificity of 99% [26], as well as by confirmatory test by RT-qPCR using Roche^®^ reagents [27] with a sensitivity and specificity of 95%.

The reporting criteria for a suspect case consisted of: (a) animals with clinical signs consistent with CSF (high fever, anorexia, gastrointestinal, symptoms, general weakness, and conjunctivitis), or (b) An epidemiological link to CSFV. Confirmed positive cases were identified by the detection of antigens of CSFV (Ag Elisa, RT-qPCR); the others with negative test results and lack of clinical signs were classified as controls (Figure 1). 

### 2.3. Questionnaire

The passive surveillance system used a questionnaire (health-event-reporting form) designed to obtain information provided by the owner of the animals and then registered online by the official veterinary; the information includes: demographic data of the owners and premises, geographic coordinates, chronology (dates of notification and follow-up), animal species, vaccination declaration, clinical signs, presumptive syndrome, collection of material, characteristics of samples, laboratory tests, animal population, animal movement and probable origin of the disease.

The information was collected by trained NVS veterinarians following the data protection procedures of Ecuadorian authorities. The information recorded throughout the country was continually monitored by the national surveillance team (headquarters), who checked the data for completeness and errors. Additional information and descriptions of the questionnaire is available on Appendix A. 

### 2.4. Multivariable Logistic Analysis 

All analyses were performed at the herd level and stratified according to CSF status (case or control). Variables were organised by type; continuous variables were transformed into dummies, setting their levels according to biological or legal cut-off points (Table 1). Descriptive statistics assessed the distribution of cases and controls. The dichotomous dependent variable used was the number of premises infected or not infected with CSF. The evaluation of individual variables of the Ecuadorian surveillance system was based on the association of each explanatory variable with the binary herd-level outcome, using univariate logistic regression [28]. We avoided case-control matching due to the potential of creating selection bias, losing precision and statistical power and not having a prior local analysis of strong well-measured confounding variables [29]. 

A multivariate logistic regression model was implemented to assess the association of explanatory variables with the outcome, using a manual forward stepwise selection [30]. We included each variable in descending order of statistical significance in the univariate models, considering a Chi-squared association to keep in the final model. For each insertion of new variables, we observed the changes in the odds ratio (OR) and the significance of each beta *βi* (Wald test), assessing them at each step. Collinearity was analysed using variance inflation factors analysis [31]. The goodness of fit of the final model was measured using the conditional R2 [32], Receiver operator curve (ROC), and Hosmer-Lemeshow goodness of fit test (GOF) (*p* > 0.05) [33]; Additionally, we used the Bonferroni outlier test and graphical analysis, looking for influential observations.

### 2.5. Spatiotemporal Bayesian Analysis

The analysis used the population and cases were restricted to 2017–2020 due to the lack of cadastral and vaccination information prior to the implementation of the official vaccination in 2017. Data were organised to contain the aggregated annual population over each parish (1040) using time-series missing value imputation [34] for areas without information. Variables were centred and scaled by dividing the centred value by the standard deviation. The variables used to fit the model were the number of CSF vaccine doses applied per km^2^, average temperature (°C) and average precipitation (mm), constructing several models. Temperature and precipitation were extracted from (https://worldclim.com/, accessed on 1 June 2022) at a spatial resolution of 2.5 arc-minutes (~5 km^2^). 

Parish vaccination coverage was adjusted considering the population and the doses applied against CSF, considering 1.55 as the average number of doses a pig receives in a calendar year, according to the average lifespan from birth to slaughter (234 days) [35]. We used penalised priority priors model complexity, specified by the divergence between a flexible model and a baseline model; to define the spatial random effect, a neighbourhood matrix from the polygon list was needed, based on regions (parishes) that share two or more boundary points. The spatiotemporal model uses the disease count *Yij* observed in area *i* and time period *j*, modelled as:(1)$$Y_{\mathrm{ij}}∼\mathrm{Po}\left(E_{\mathrm{ij}}\theta_{\mathrm{ij}}\right);i=1,\ldots N;j=t1,\ldots ,\mathrm{tN}$$
where $E_{\mathrm{ij}}$ is the expected number of cases and $\theta_{\mathrm{ij}}$ is the relative risk, both in the given area (*i*) and time period (*t*) (Equation (1)). Three sets of components for $\log\left(\theta_{\mathrm{ij}}\right)$ were considered.
(2)$$\log\left(\theta_{\mathrm{ij}}\right){=\alpha +u}_{i}{+v}_{i}$$
where alpha represents an overall risk in the study region, *ui* is the correlated heterogeneity, which models the spatial dependence between the relative risks, and *vi* is the unstructured exchangeable component that models uncorrelated noise (Equation (2)).
(3)$$\log\left(\theta_{\mathrm{ij}}\right){=\alpha }_{0}{+A}_{i}{+B}_{j}{+C}_{\mathrm{ij}}+\mathrm{var}1+\mathrm{var}2+\mathrm{var}\ldots n$$
where *Ai* represents the spatial group, *Bj* is the temporal group, and *Cij* is the space–time interaction group (*Ai* = *ui* + *vi*) using the most popular model to spatially define disease, also known as Besag–York–Mollié (BYM) [36], where the clustering component *ui* is modelled with the conditional autoregressive distribution (CAR) [37], smoothing the data when two areas share a common boundary given by the neighbourhood matrix (*Bj* = *βtj*). Using an independent and identically distributed Gaussian random effect (iid). (*Cij* = *δitj*), where *ui* + *vi* is an area random effect, *βtj* is a linear trend term in time *tj*, and *δitj* is an interaction random effect between area and time (Equation (3)) [38]. 

To evaluate the models, we used the deviance information criterion (DIC) and the posterior predictive *p* value. To suggest a parish that required a priority of care, we used the priority index (PI) which is a risk-based percentage scale that ranks the units of analysis, given by the fitted effects weighted by their probability and a cut-off value [39]. The models were implemented using the integrated nested laplace approximation (INLA) [40]. We used choropleth maps to represent the spatiotemporal distribution of the population, observed cases, expected cases, infection risk (relative risk) and priority index.

All analyses were run in R V4.2.0 (https://cran.r-project.org/, accessed on 1 May 2022), and the following packages were used: Tydiverse [41] for data preparation, ‘car’ [42], ‘stats’, ResourceSelection’ [43], and ‘modEvA’ [44] to run the logistic models. For spatiotemporal analysis, we used TSImpute [34], INLA to compute the Bayesian inference [45], and INLA-outputs and rgdal [46] for geographic plotting.

## 3. Results

### 3.1. Descriptive Analysis of the Variables Influencing the Risk

The full dataset contained 63 variables, most of which were used for administrative purposes. Fifteen variables selected for the univariate analysis consisted of six dichotomous, six nominal and three continuous variables. They were then classified into the population level (*n* = 4), herd level (*n* = 9) and animal level (*n* = 2). Time until notification was transformed considering a cut point of 7 days (one week) since the onset of clinical signs. The pigs’ population on the premise was transformed, considering cut points of 25 and 190 pigs, and considering percentiles 75 and 95 of the national population due to previous national limits for backyard and commercial premises. The age of the animals considered cut-off points of 2 and 6 months due to the official CSF vaccination recommendation: first dose is applied after 45 days and revaccination occurs at 180 days of age (Table 1).

The average time for official notification to the NVS was more than one week (9.3 days) for cases and one week (7.0 days) for controls. The median premise population was similar for cases (13.5 pigs) and controls. The mean age of pigs was similar for cases and controls (~5 months) (Table 2).

Farmer vaccination declaration was higher for case herds (71%) than for control herds (6%). Recording of vaccination (based on official records) was lower for cases than for controls. Both cases and controls had a high percentage of swill feed use. The entry of animals within the last 30 days happened in 39% of cases and 22% of controls. Only 5% of the cases and 4% of controls have other species on the property (Table 3).

Historical case presentation decreased over the years, with the highest proportion of cases (48%) occurring between 2014 and 2015 and the lowest (14%) occurring between 2019 and 2020. The proportion of cases (43%) and controls (45%) was higher in the highlands. More than half of the case notifications (51%) were reported by the owner, followed by health sensors (42%), which are volunteers selected by the NVS to enhance the surveillance system directly from communities across the country. There was a higher proportion of cases in the third community network, known for their high density of backyard producers and indigenous communities [47], located in the centre of the country. The highest proportion of cases number was in commercial production (Table 3).

### 3.2. Description of Cases and Controls

The surveillance database contained 1254 questionnaires, 338 of which were confirmed CSF cases. The premise categories were 50.79% family (637), followed by 28.39% commercial (356), 17.98% backyard (224) and 0.03% industrial (37); the distribution of cases and controls over time is illustrated in Figure 2. The highest case presentation corresponded to October 2015 with 14 monthly cases, followed by March 2014 with 13 cases, and the lowest corresponded to 2020 with ≤2 monthly cases.

Distributed by months, the mean number of cases was 4.39 ± 3.09; with controls, the mean was 11.31 ± 6.38 There was a tendency to increase twice a year around April and October (Figure 3).

### 3.3. Multivariable Logistic Analysis

We ran 15 univariable models; twelve of the assessed variables were associated with CSF (*p* < 0.20). We found a paradoxical fit (Type III error) opposite of the true effect [48], produced by CSF vaccination declaration, giving an incorrect direction of association and increasing the odds when the farmer declares vaccination (38.67 OR), instead of the expected protective effect conferred by the vaccine. The administrative region showed a higher risk in the coastal region; as this is associated with spatial dependence, we retired from the model and included it in the spatiotemporal analysis. The univariable analysis is presented in Table 3. 

During the stepwise forward selection of variables, we evaluated eight models; thus, variables that were not statistically significant (*p* > 0.05) had an incorrect direction of association (self-declaration of vaccination) or that were at the animal level (age of animals) were excluded. Variables that showed a strong association (*p* < 0.0001) in the univariable model maintained their individual and model significance when adjusted in the final multivariable model (Table 4).

Factors that substantially increased the odds of CSF occurrence at the herd level were swill feeding (OR 8.53), time until notification (OR 2.44), entry of animals in the last 30 days (OR 2.01) and lack of CSF vaccination (OR 1.82) (Table 4). The final logistic model presented good fit (GOF = 0.99, AUC = 0.72). Individual collinearity diagnostics for each variable resulted in individual GVIFs below 1.062. There was no outlier with a significant influence on model fitting, according to the Bonferroni outlier test (*p* = 0.006); also, there was no correlation between residuals. Details of the manual stepwise model construction, analysis and the predicted probability map of occurrence aggregated by parishes is available in Appendix A.

### 3.4. Spatiotemporal Descriptive Analysis

The Ecuadorian administrative division has 1040 parishes; 16 were removed because of them being islands and 18 were located in the Amazon rainforest with no record of domestic pigs. The length of the neighbouring areas was 4024 (1006 for each year), with 271 (6.49%) imputed gaps. When comparing the imputed data with the original dataset, they were not significantly different (t-test: *p* = 0.55). The final neighbour list contained 1006 parishes with an average of 5.71 parish neighbours. The average parish area was 198.03 ± 249.48 km^2^, with a range from 2.23 to 2429.64 km^2^. The annual average of registered premises was 115,411.8, housing an average of 1,633,922 pigs. 

The annual average of CSF vaccine doses was 2.4 million, and average vaccine doses per square kilometre increased from 16 to 23 (2017–2019); the highest average vaccination coverage was 81% in 2019. The number of applied doses increased from 1.8 million in 2017 to 2.4 million in 2020. The annual average of the premises was 115,411 (Table 5).

What stands out in Figure 4 is the decline of the number of observed cases, corresponding to 39 in 2017, 60 in 2018, 33 in 2019 and 13 in 2020; it is possible to observe a significant reduction in the number of cases especially in the highlands over the years; the expected cases figure is available in Appendix A.

Figure 5 reveals that there was a marked higher density on the number of doses of CSF applied by square kilometre in the western centre (Santo Domingo), the north (Carchi), west south (El Oro) and the central highlands (Cotopaxi, Chimborazo), and this general pattern repeated over the years; however, there were 105, 78, 52 and 62 parishes without vaccination coverage in 2017, 2018, 2019 and 2020, respectively (note the white parishes on Figure 5).

Temperature and precipitation in Ecuador are modulated by the Andes mountains, warmer temperatures in the Amazon and Coastal regions and cooler temperatures in the highlands. There is a range difference of 11 °C in the coastal regions, 20.5 °C in the highlands and 18.9 °C in the Amazon. Precipitation on the Amazon is almost three times higher than the highlands and two times compared with the Coastal (Table 6).

### 3.5. Spatiotemporal Relative Risk

The annual average relative risk dropped from 4.01 in 2017 to 1.30 in 2020. Regarding the doses of vaccine applied per km^2^, they behaved as an expected protective factor, which means that an increase in one SD in the doses applied per kilometre decreased the risk by 33%. Temperature was a risk factor, considering that an increase in one SD in temperature degree increased the risk by 16.7%. Precipitation had no effect: RR = 1.00 (1.00–1.001) (Table 7).

The proportion of variance explained by each component was 57% for the random effect (iid), a major contributor to the explained variance, and 43% for the spatiotemporal (besag). The DIC mean deviance was 1140 and the effective number of parameters was 136.7. 

The spatial distribution of risk at parish scale is shown in Figure 6; hot spots of increased risk were spatially identified on the map, in the coastal southwestern and also the south-eastern Amazon, reducing the risk over the years. Locations with a higher risk are visually located in neglected parishes specially in the Amazon. The posterior distributions of the covariates of the spatiotemporal model, as well as the annual average relative risk, show a reduction during the study period (Appendix A).

According to the priority index (PI), primary parishes of concern include *Tundayme* located in the eastern Amazon, followed by *Tachina* in the north-western and *Paletilla* in the southern zone (Figure 7). The provinces with higher risk, considering the average RR per province in the year 2020, were *Morona Santiago* (3.68), *Los Rios* (3.12) and *Santa Elena*, (3.07); the average relative risk intended to prioritise prevention and control activities by province is available in Appendix A, and the individual parish RR details are in Appendix A. Considerations about some of the limitations of the study are available in the Appendix B.

## 4. Discussion

When countries mount resource-intensive control strategies for a high-impact disease such as CSF but fail to reach the goal of control and elimination, a deeper analysis of the disease dynamics and the implemented control interventions is needed to identify strategic intervention points. Despite the fact that, in general terms, many CSF risk factors are known, their relevance in the specific setting of a pig sector and the respective control program is ideally assessed using all available data.

Swill feeding is one of the main risk factors for CSF transmission; it is common and rooted in the cultural tradition of backyard producers [49]. Therefore, it is very likely to be a key disease driver in endemic areas of Andean countries. The Agricultural Health Law of 2019 [50] established best practices for animal feed, but lacked specific regulations on swill. Consideration needs to be given to promoting risk-reducing practices such as heat treatment [51] and stricter regulations that prohibit the use of animal protein as a feed source for pigs.

Vaccination misreporting could be related to the producer’s lack of knowledge regarding veterinary treatments linked with injections (vaccination, iron supplementation in piglets, deworming or other). Fear associated with owners’ legal responsibilities and misunderstandings during the interviews may also lead to misreporting [52]. In Indonesia, vaccination against CSF resulted in an increased risk of CSF due to inaccurate vaccination claims [24]; considering these facts, reporting behaviour could be further analysed as an early target of the surveillance programme [53], suggesting that communication and health education activities might be advisable to improve producers’ understanding of animal disease prevention and control practices.

The increased risk of pigs at the age of 3–6 months (univariate model) could be related to the fact that young animals may be more exposed to CSFV because this is the age at which, in rural communities, they are moved to animal markets, using various means of transport. Additionally, this reflects a complicated age from the immunological perspective because maternal immunity fades out after three months [54], and animals not vaccinated become susceptible just as animals vaccinated too early, where maternal antibodies interfere with the vaccination. Nowadays, the established recommendation for piglets first vaccination in Ecuador is at relatively early, 45 days, and revaccination is recommended every 6 months. This practice might have to reassessed once targeted sero-surveillance studies are conducted to clarify the effects of vaccination ages and herd immune status. 

Maternal-derived antibody (MDA) interference is the most common factor affecting the induction of protective immunity against CSFV [54]; in Thailand, the vaccination program has been implemented for decades without achieving eradication [55]. In addition, emergency vaccination protocols implemented in very young piglets, especially during an outbreak, could be further analysed. It would be necessary to evaluate diagnostic tools (rapid test) [56] that could detect non-clinical, persistent CSF forms in the field, as well as apply vaccination serological monitoring tools [57]. 

Some cases occurred on premises with previous vaccination records, possibly related to illegal movements of unvaccinated animals [47] or vaccination failures because of poor handling and mismanagement, as evidenced in neighbouring Colombia [4,20]. The risk associated with higher temperatures in the spatiotemporal analysis could also be linked to problems in the vaccine cold-chain in regions such as the Amazon and the coast, which represent a logistical challenge when addressing average temperatures above 20 degrees Celsius, corroborating that vaccination strategies alone may not eradicate the disease [58,59]; Furthermore, possible antigenic alterations because of vaccination pressure and their effects on the epidemiology of the disease could also be further analysed [60,61].

The identified individual parish risk could help identify neglected territories; as in many developing countries with limited resources for disease control, prioritisation is often done on the basis of historic surveillance information. Therefore, reduced surveillance sensitivity may leave areas of high risk unnoticed. Our spatial model included all parishes and considered the influence of their neighbours to improve the predictions [62]. Concepts such as spatial RR or excess risk might be difficult to interpret outside the scientific community, but the priority index (PI) could facilitate the understanding and communication of which parishes should be prioritised.

The identification of the risk factors should respond to the initial demand of the NVS and contribute to the implementation of a risk-based surveillance strategy for animal diseases. As risk factors are specific for each disease, new studies could be implemented using depurated data and methodology for prevalent diseases where their symptomatology could be confused with CSF, such as the porcine reproductive and respiratory syndrome [63,64], as well as prepare the surveillance system for re-emerging diseases such as African swine fever, currently detected in Central America [65,66].

## 5. Conclusions

The purpose of the current study was to determine the risk factors and spatiotemporal implications associated with CSF in Ecuador. The results show once again the complexity that the CSF control program is facing, particularly if the pig sector is diverse and comprises a large share of premises falling under the backyard category. Here, NVS faces risky production methods combined with a reduced knowledge of disease prevention and compliance with sanitary regulations.

One of the more significant findings from this study is the identification of swill feeding, time until notification, introduction of new pigs and the lack of vaccination, as the risk factors; the second major finding is the parish priority index based on spatiotemporal risk. We hope these results would be useful to improve intervention focusing on specific parishes countrywide.

The results highlight the importance of improving the overall surveillance system and the need for better methods to inform decision-makers and stakeholders.

## Figures and Tables

**Figure 1 viruses-15-00288-f001:**
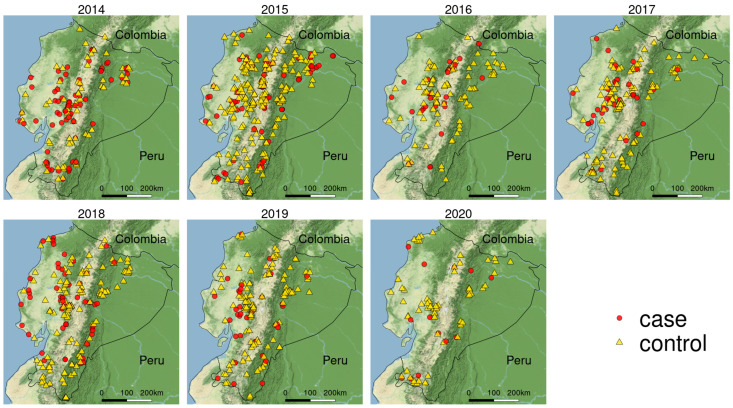
Spatial representation of the study area and location of cases and controls of CSF in Ecuador study period 2014–2020.

**Figure 2 viruses-15-00288-f002:**
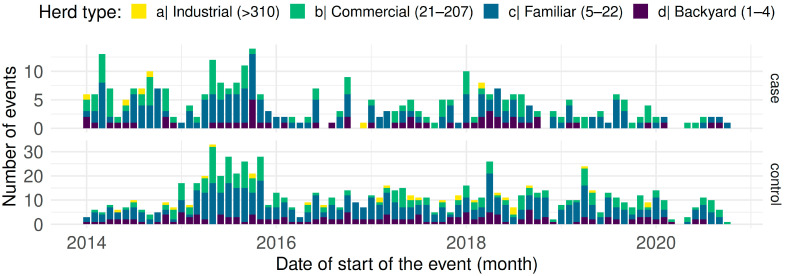
Distribution of events (cases and controls) reported between 2014 and 2020 in Ecuador; bars represent monthly counts, height and colours of bars according to the type of herd (different scales on the y-axis).

**Figure 3 viruses-15-00288-f003:**
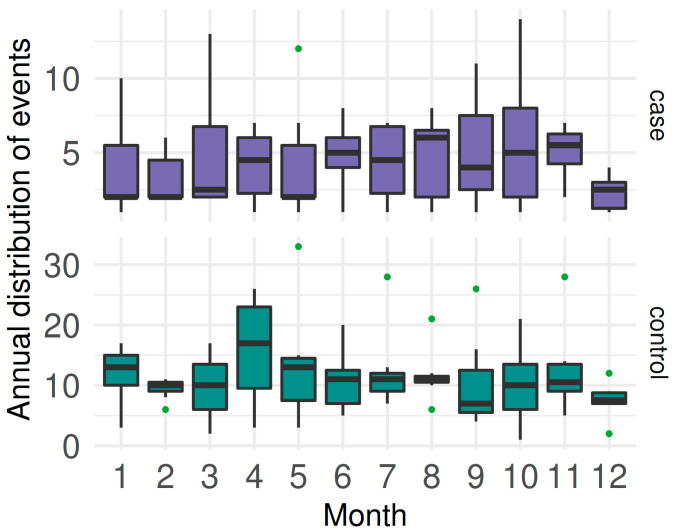
Boxplots of monthly distribution of CSF cases and controls from 2014 to 2020 in Ecuador (different scales on the y-axis).

**Figure 4 viruses-15-00288-f004:**
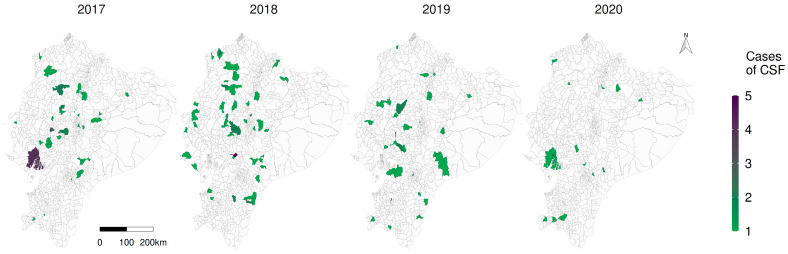
Representation of the number of observed CSF cases (number of positive premises) in Ecuador grouped by parish from 2017 to 2020.

**Figure 5 viruses-15-00288-f005:**
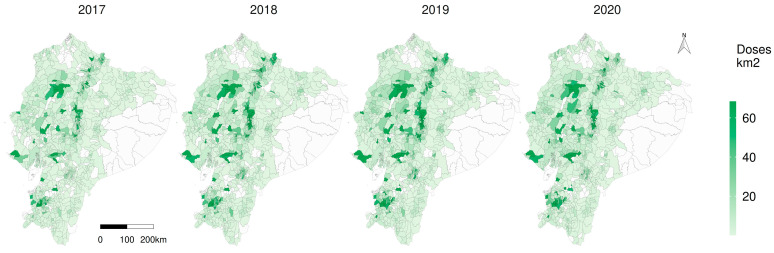
Classical swine fever vaccination density per square kilometre in Ecuador. Polygons of the parishes on the map. White parish without applied doses.

**Figure 6 viruses-15-00288-f006:**
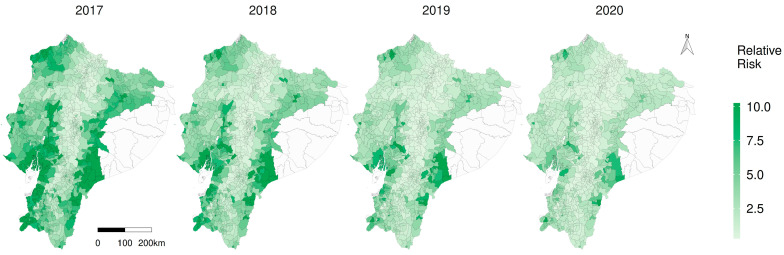
Spatiotemporal representation of the relative risk (RR) of CSF in Ecuador.

**Figure 7 viruses-15-00288-f007:**
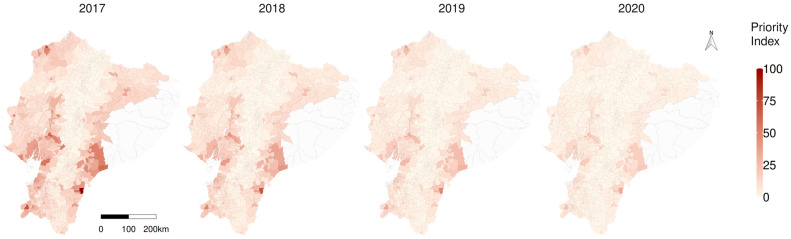
Spatiotemporal representation of the priority index (PI) to fight classical swine fever in Ecuador.

**Table 1 viruses-15-00288-t001:** Description of variables influencing risk and their levels. Data available in Ecuador’s CSF surveillance system from 2014 to 2020; levels grouped by risk characterisation.

Factors Influencing Risk	Description of Variables Captured by the Surveillance System	Category
^†^ Control program	Active national control program (vaccination and mobilisation control) on the movement of interview.	Dichotomous ^¶^
^†^ Network community	Community to which the premise belongs according to its parish location (42).	5 communities
^†^ Year	Year of the event.	2014–2020
^‡^ Introduction of new pigs	Reception of pigs within 30 days of onset of clinical signs.	Dichotomous ^¶^
^‡^ Administrative region	Region of the premise according with the 24 administrative provincial division.	Amazon, coastal, highlands
^‡^ Time until notification	Number of days from onset of clinical signs to notification to the NVS.	0–7, >7
^‡^ Other species	Existence of species other than swine in the premise.	Dichotomous
^‡^ Premise population	Number of pigs on the premise.	1–25, 26–189, >190,
^‡^ Swill feeding	Evidence of feeding pigs with swill feed, home-made leftovers.	Dichotomous ^¶^
^‡^ Type of premise	Classification of premises according to production category.	Backyard, Family, Commercial, Industrial
^‡^ CSF vaccination declaration	Owner’s declaration of vaccination against CSF in its premise.	Dichotomous ^¶^
^‡^ CSF vaccination record	Official record of vaccination against CSF within the last 180 days.	Dichotomous ^¶^
^‡^ Who makes the notification	Person who contacted the NVS to make the notification.	Owner, NVS, Sensor
^§^ Age	Age in months of the first suspected animal on the premise, also the very first sampled.	1–2, 3–6, >=7
^§^ Breed	Breed of the animals on the premise.	Landrace (white), Indigenous (black).

Levels of risk characterisation: ^†^ Population level, ^‡^ Herd level, ^§^ Animal level. ^¶^ Dichotomous: having only two possible values, 0 = No, 1 = Yes.

**Table 2 viruses-15-00288-t002:** Descriptive measures of continuous variables from the 2014–2020 CSF risk factor analysis in Ecuador.

	Case Herds (*n* = 338)			Control Herds (*n* = 916)		
	Mean ± SD	Median (Q2,Q)	Range	Mean ± SD	Median (Q2,Q)	Range
Time until notification	9.29 ± 9.13	7 (3–13)	0–70	7.00 ± 13.37	3 (2–7)	0–201
Premise population	38.54 ± 101.95	13 (6–33)	1–1323	125.4 ± 893.54	13 (6–27)	1–13,804
Age (months)	5.06 ± 5.54	3 (2–5)	1–48	5.89 ± 7.69	3 (2–5)	1–72

**Table 3 viruses-15-00288-t003:** Results of univariable logistic regression analyses, to assess associations of CSF during 2014–2020 in Ecuadorian swine herds. Variables are ordered by their level of significance.

Variable	Category	Total	Cases (%)	OR (Crude)	95% CI
CSF vaccination declaration	No	959	98 (0.1)	1	
	Yes	295	240 (0.81)	38.34 ***	(26.75–54.94)
Swill feeding	No	216	13 (0.06)	1	
	Yes	1038	325 (0.31)	7.12 ***	(4.16–13.29)
Time until notification	7 days	882	191 (0.22)	1	
	>7 days	372	147 (0.4)	2.36 ***	(1.82–3.07)
Introduction of new pigs (last 30 days)	No	920	207 (0.22)	1	
	Yes	334	131 (0.39)	2.22 ***	(1.7–2.91)
CSF vaccination record	Yes	710	157 (0.22)	1	
	No	544	181 (0.33)	1.76 ***	(1.37–2.26)
Administrative Region	Highlands	558	146 (0.26)	1	
	Coastal	311	112 (0.36)	1.59 **	(1.18–2.14)
	Amazon	385	80 (0.21)	0.74 ^‡^	(0.54–1.01)
Year	2019–2020	244	48 (0.2)	1	
	2016–2018	526	129 (0.25)	1.33	(0.90–1.97)
	2014–2015	484	161 (0.33)	2.03 ***	(1.39–3.01)
Age (months)	1–2	426	93 (0.22)	1	
	3–6	588	186 (0.32)	1.68	(1.25–2.27)
	>=7	248	60 (0.24)	1.14	(0.77–1.68)
Control Program	No	609	190 (0.31)	1	
	Yes	645	148 (0.23)	0.66 **	(0.51–0.84)
Who does the notification	Owner	736	171 (0.23)	1	
	NVS	86	25 (0.29)	1.35	(0.79–2.27)
	Sensor	432	142 (0.33)	1.62 ***	(1.23–2.12)
Premise population	>190	53	8 (0.15)	1	
	1–25	910	232 (0.25)	1.92 ^‡^	(0.88–4.8)
	26–189	293	99 (0.34)	2.86 **	(1.27–7.31)
Network Community	1	219	53 (0.24)	1	
	2	157	57 (0.36)	1.78 *	(1.11–2.87)
	3	359	102 (0.28)	1.24	(0.83–1.87)
	4	184	49 (0.27)	1.14	(0.71–1.83)
	5	335	77 (0.23)	0.93	(0.62–1.43)
Breed	Indigenous black	93	20 (0.22)	1	
	Landrace	970	260 (0.27)	1.33 †	(0.79–2.36)
Other species in the premise	No	1205	322 (0.27)	1	
	Yes	51	17 (0.33)	1.37 †	(0.76–2.49)
Type of premise	Industrial	37	6 (0)	1	
	Commercial	356	106 (0.08)	2.19 ^‡^	(0.86–6.6)
	Family	637	169 (0.13)	1.86	(0.75–5.56)
	Backyard	224	57 (0.05)	1.76 †	(0.68–5.53)

† Indicates an association *p* > 0.20, these variables were excluded in the multivariable models. Signif.: *** *p* < 0.001, ** *p* < 0.01, * *p* < 0.05, ^‡^
*p* < 0.1. CI: Confidence interval.

**Table 4 viruses-15-00288-t004:** Multivariable logistic regression model assessing the association of variables with the odds of CSF between 2014 and 2020 in Ecuador.

Variable	Category	Estimate	SE	OR (95% CI)
Intercept		−3.69	0.31	
Swill feeding	No	-	-	1
	Yes	2.14	0.30	8.53 (4.92–16.11) ***
Time until notification	1–7 days	-	-	1
	>7 days	0.89	0.14	2.44 (1.84–3.23) ***
Introduction of new pigs (last 30 days)	No	-	-	1
	Yes	0.69	0.15	2.01 (1.51–2.67) ***
Vaccination record CSF	Yes	-	-	1
	No	0.59	0.14	1.82 (1.39–2.38) ***

Model performance metrics: Chi-sqrt: >0.001, GOF Hosmer–Lemeshow: 0.99, AUC: 0.72, D2: 0.11, R2: 0.13. Significance: *** 0.001. Level of risk characterisation: Herd level.

**Table 5 viruses-15-00288-t005:** Centrality measures of model variables (fixed effects) aggregated by parish distribution in Ecuador.

	2017		2018		2019		2020	
Variable	Average	Median (max)	Average	Median (max)	Average	Median (max)	Average	Median (max)
Doses CSF/km^2^	15.74	2.31 (976.5)	23.78	4.15 (1057.6)	27.63	5.08 (1282)	23.0	4.2 (1477.3)
Population of pigs	1671.3	284 (224,448)	1948.7	391 (256,107)	2030.4	447 (254,042)	1867.1	434.5 (227,186)
Vaccine coverage %	60	64 (129)	71	80 (108)	81	100 (105)	70	73 (103)

**Table 6 viruses-15-00288-t006:** Descriptive measures of covariants of the spatiotemporal model in Ecuador.

	Coastal	Highlands	Amazon
Covariate	Mean ± SD (range)	Mean ± SD (range)	Mean ± SD (range)
Doses vac. km^2^	18.93 ± 69.83 (0–636.9)	19 ± 61.01 (0–976.5)	1.77 ± 2.99 (0–15.26)
Temperature (°C)	24.36 ± 1.63 (15–26)	14.26 ± 4.43 (4.6–25.1)	20.60 ± 4.19 (6.8–25.7)
Precipitation (mm)	1362.24 ± 712.20 (122–3253)	1012.89 ± 434.16 (432–3824)	2718.53 ± 957.47 (722–4482)

**Table 7 viruses-15-00288-t007:** Summary fixed effects of covariates on the estimated risk (RR) of CSF in a spatiotemporal Bayesian model in Ecuador 2017–2020.

Covariate	Univariate			Multivariate			Relative Risk	
	Mean	(0.95 % CI)	DIC	Mean	(0.95 % CI)	DIC	RR	(0.95 % CI)
Intercept	–	–	–	−2.26	(−3.30, −1.31)	1140	–	–
Time (years)	–	–	–	−0.36	(−0.51, −0.21)	–	0.70	(0.60, 0.81)
Doses by km^2^	−0.309	(−0.68, 0.02)	1188	−0.41	(−0.77, −0.09)	–	0.67	(0.46, 0.91)
Temperature	0.158	(0.11, 0.21)	1140	0.15	(0.11, 0.20)	–	1.17	(1.12, 1.22)
Precipitation	0.001	(0.00, 0.001)	1156					
Adm. Reg. Coastal	0.822	(0.04, 1.57)	1160					
Adm Reg. Amazon	1.95	(1.14, 2.76)						

Adm. Reg.= Administrative region.

## Data Availability

The data are not publicly available due to “The surveillance data contain private information of each Ecuadorian pig farmer and therefore cannot be made available due to legal restrictions”.

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
