# Peer review of "Risk Factors and Spatiotemporal Analysis of Classical Swine Fever in Ecuador"

_viruses, 2023, doi:10.3390/v15020288_

Round 1

Reviewer 1 Report

"Risk Factors and Spatiotemporal Analysis of Classical Swine Fever in Ecuador" aims to determine the risk factors and spatiotemporal implications associated with CSF in Ecuador and to identify the most at risk locations.

The study is interesting and provides useful data for the risk analysis on the diffusion of classical swine fever (CSF) in Ecuador. The information included in this paper is important in order to address future CSF control programmes and to improve the overall surveillance system in this country. 

The study design is well performed and is based on a herd-level case-control study using a logistic and spatiotemporal Bayesian model.

Even if the study can be improved by obtaining more precise and consistent data in the future, I think that the manuscript is worthy of publication.

Line 19: Spatiotemporal with lowercase letter

Line 93: Delete one excess space.

Lines 107-111: Is not clear when you used ELISA test and when RT-qPCR. Why did you use 2 different methods? Can you explain this?

Line 122: Delete one excess space.

Line 156: Correct as "data were organized"

Lines 161 and 163: Write Km2 with superscript number

Table 1: It would be good to put it all in one page and not divided. Can you better explain the meaning of "dichotomous" in this table?

Line 419: "We" with lowercase letter

Author Response

Response to Reviewer 1

We would like to specially thank the reviewer for the time invested in the improvement of the article.

Line 19: Spatiotemporal with lowercase letter

Response 1: Changed accordingly in line 19.

Line 93: Delete one excess space.

Response 2: Changed accordingly in line 84.

Lines 107-111: Is not clear when you used ELISA test and when RT-qPCR. Why did you use 2 different methods? Can you explain this?

Response 3: We agree with the reviewer that this important detail needs to be specified. We included that the screening method was the ELISA and the confirmation method used rt-PCR, in lines 97 and 98.

Line 122: Delete one excess space.

Response 4: Changed accordingly in line 107.

Line 156: Correct as "data were organized"

Response 5: Corrected accordingly in line 136.

Lines 161 and 163: Write Km2 with superscript number

Response 6: Corrected accordingly in line 139,141.

Table 1: It would be good to put it all in one page and not divided. Can you better explain the meaning of "dichotomous" in this table?

Response 7: We included a page break on the table and in the foot notes of the table added the dichotomous definition.

Line 419: "We" with lowercase letter

Response 7: Corrected accordingly in line 350.

Reviewer 2 Report

I have reviewed the draft of the article and have reached the conclusion that this is, in its present form, suitable with minor revision for publication in “Viruses”.  The resolution of maps should be increased.

This article presents an original study  to determine the risk factors and spatiotemporal implications associated with CSF in Ecuador. However, I think it would be very useful for an academic to review this paper, especially if they are interested in the logistic and spatiotemporal Bayesian model. 

Author Response

Response to Reviewer 2

We would like to specially thank the reviewer for the time invested in the improvement of the article.

The resolution of maps should be increased.

Response 1: We agree with the reviewer about the low images resolution in the PDF. Full resolution tiff images (8503 x 2834), were provided accordingly.
